# Dynamics of Confined Short-Chain *alkanol* in *MCM-41* by Dielectric Spectroscopy: Effects of *matrix* and *system* Treatments and Filling Factor

**DOI:** 10.3390/polym12030610

**Published:** 2020-03-07

**Authors:** Josef Bartoš, Silvia Arrese-Igor, Helena Švajdlenková, Angela Kleinová, Angel Alegría

**Affiliations:** 1Polymer Institute of SAS, Dúbravská cesta, 9 845 41 Bratislava, Slovakia; Helena.Svajdlenkova@savba.sk (H.Š.); Angela.Kleinova@savba.sk (A.K.); 2Centro de Física de Materiales, CSIC-UPV/EHU, Paseo Manuel de Lardizabal 5, 20018 San Sebastián, Spain; Silvia.Arreseigor@Ehu.Eus (S.A.-I.); Angel.Alegria@ehu.eus (A.A.); 3Departamento de Física de Materiales, UPV/EHU, Apartado 1072, 20080 San Sebastián, Spain

**Keywords:** *short-chain alkanol*, *MCM-41 matrix*, relaxation dynamics, BDS, *humidity*, FT M-IR

## Abstract

The dynamics of *n-propanol* confined in regular *MCM-41*
*matrix* with the pore size *D*_pore_ = 40 Å, under various matrix conditioning and sample confining conditions, using broadband dielectric spectroscopy (BDS), is reported. First, various drying procedures with the capacitor filling under *air* or *N_2_* influence the BDS spectra of the empty *MCM-41* and the confined *n-PrOH/MCM-41 systems*, but have a little effect on the maximum relaxation time of the main process. Finally, various filling factors of *n-PrOH*
*medium* in the optimally treated *MCM-41*
*system* lead to unimodal or bimodal spectra interpreted in terms of the two *distinct* dynamic phases in the confined states.

## 1. Introduction

The topic of bulk vs. confined organic *materials* is usually studied by a variety of classic experimental techniques. Changes in dynamics of various organic *media* as reflected by specific *intrinsic* probes of material, such as dynamic density fluctuations, reorientations of magnetic and electric dipoles are detected by means of neutron scattering (NS), nuclear magnetic resonance (NMR) or dielectric relaxation spectroscopy (BDS), respectively, are often observed (see general and special reviews [1,2,3,4,5,6,7,8,9]). These are related to phase transitions and eventually, to a formation of new phase(s) induced in confined organic *medium* (*filler*) by spatial restriction or/and by wall surface of confining inorganic *matrix* (*confiner*) [1,2,3,4] with their complex impacts into dynamic properties [5,6,7,8,9]. The overall confinement effect, as seen by these standard dynamic techniques, is considered to be a result of the complex mutual interplay of the following two main factors: (i) restricting geometric effects of the pores in a given *matrix* on the investigated *medium* and (ii) the mutual interaction effects of the *medium’s entities* with the pore surface wall of the *matrix* [1,2,3,4,5,6,7,8,9].

The bulk vs. confinement topic as the multiparameter problem includes a complex mutual action of many *internal* and *external* parameters. The most studied group of *internal* parameters concerns the confining *matrix,* such as pore size, pore size distribution, pore topology (mutual pore separation or interconnectivity) and pore surface composition and the confined organic *medium,* such as its size, shape and polarity, and proticity of the *molecules*. The group of *external* parameters includes those related to the *matrix* and confined *system* conditioning and preparation, such as thermal/time treatments of inorganic *matrix* in certain atmosphere, filling procedure of confined *systems*, again under a certain atmosphere, as well as degree of filling the pores (filling factor), of the previous specifically treated *matrix* with a given *medium*. One of the most often used model inorganic *matrices* based on *silica* such as irregular *mesoporous silica gels* (*SG*) and regular *periodic mesoporous silicas* (*PMS*), e.g., *MCM-41* [10,11,12] and *SBA-15* [13,14], has the hydrophilic character due to the presence of three basic kinds of polar *silanol groups*: isolated (free) *silanol* [*O≡Si-OH*], geminal *silanol* or *silanediol* [*O=Si=*(*OH*)*_2_*] and vicinal *silanol* [*O≡Si-OH...HO-Si≡O*], with their different adsorption ability to the environmental *moisture* (*H_2_O* from *air*) or other *agents* [15,16,17,18] and the capability of having various modification treatments [17,18]. Consequently, many BDS works addressed this important *water*-adsorption phenomenon in various *silicas* [19,20,21,22]. However, in spite of the importance of these *external* parameters, a relatively small amount of attention was devoted to *systematic* studies of their variations from the viewpoint of their impact into the dynamic response of the various confined *organics* [23,24]. This is also evident from the frequent absence of *detailed* descriptions of the empty *matrix* and confined *system* conditioning, as well as of the filling conditions during the preparation of the confined *systems.*

The knowledge situation is further complicated for the special group of *H-bonded compounds*, a very important class of organic *media*. Although the simpler *poly alcohols,* such as *1,2-ethanediol, ethylene glycol* (*EG*), *1,2-propanediol*, *propylene glycol* (*PG*) and *1,2,3-propanetriol, glycerol* (*GL*) were among the first model *organic media* investigated after insertion into various inorganic *silica matrices* by BDS [25,26,27,28,29,30], BDS works on confined *monohydroxy alcohols* are rather limited [24,31,32,33]. This family of *alcohols* exhibits basically two peak features, where the main peak consists of two strongly superimposed subpeaks from the dominating Debye relaxation and the essentially less intense primary *α* one together with the well-separated secondary *β* process [34]. First, branched longer *monohydroxy alkanols* were studied where well-separated Debye- and *α*-processes were observed [31,32]. Later the first simplest members of the *n-alkanol family*, i.e., *methanol* and *ethanol,* were measured under confinement in *MCM-41*, showing a single global main peak, without any resolution of possible individual processes [24,33].

*N-propanol, CH*_3_*CH*_2_*CH*_2_*OH,* (*n-PrOH*) is a typical amphiphilic organic *substance* containing both the *apolar* alkyl group and the *polar* hydroxyl one in *the molecule*, with its ability to form intermolecular *H*-bonded *associates* (*aggregates*) in the liquid state [34]. Basic structural and dynamic properties of *n-PrOH* in the bulk state were investigated by classic structural X-ray and neutron diffraction (XD and ND) techniques, together with various Monte Carlo and molecular dynamics (MC and MD) simulations, as well as Empirical Potential Structure Refinement (EPSR) modeling [35,36,37,38], and by dynamics techniques, such as BDS and neutron scattering (NS) [39,40,41], so that the structural–dynamic state of the bulk *liquid n-PrOH* is quite well understood. On the other hand, a few works only were focused on dynamic behavior of small *alkanols*, such as *n-PrOH* confined in various microscopic porous inorganic *matrices* [42]. In addition, recently, some of us performed electron spin resonance (ESR) studies via *spin probe* method on the bulk *n-PrOH* [43] and on the respective saturated confined states in the irregular virgin *silica gels* (*SGs*) [44], as well as in the regular virgin *MCM-41* [45], showing significant differences in the *spin probe 2,2,6,6-tetramethyl-piperidinyl-1-oxy* (*TEMPO*) dynamics, as well as in its dynamic heterogeneity with respect to the bulk *n-PrOH medium*.

In this contribution, we present the results of detailed systematic BDS studies of the relaxation dynamics of *n-PrOH* embedded in a periodic *silica*-based *matrix*. The *MCM-41 type silica matrix* with the defined pore size, pore topology and pore composition [10,11,12] was investigated in detail by a variation of *external* variables, such as (i) thermal, i.e., temperature/time treatment and capacitor filling of the empty *MCM-41 matrix* under various atmospheres, i.e., *air* and *nitrogen*; (ii) pore and capacitor filling of the *n-PrOH/MCM-41 systems* under various atmospheres; and (iii) different filling factor of the confined *medium/matrix system* for the most optimally treated empty *MCM-41 matrix*, as well as the most optimally prepared confined *n-PrOH/MCM-41 system*.

## 2. Materials and Methods

Anhydrous *n-Propanol* (*n-PrOH*) from Sigma-Aldrich, Inc., Germany, with a purity of 99.7% was used as confined organic *medium* (*filler*). Regular virgin *Mobil Composition of Matter* (*MCM-41*) *silica matrix* having regularly (parallelly) ordered cylinder-like channels with the mean pore size *D*_pore_ = 40 Å, from Sigma-Aldrich, Inc., Germany, was utilized as a confining inorganic *material.*

### 2.1. FTIR Measurements

First, the conditioning by *drying* procedure of the empty *MCM-41* was monitored by Fourier transform middle-infrared (FT M-IR) spectroscopy. The empty *MCM-41 matrix* was undergone to stepwise *drying* for 0, 1, 3, 6 and 7 days, in the vacuum oven (VO), and subsequently immediately measured in FTIR spectrometer Nicolet 8700^TM^, Thermo Scientific, Madison, WI, USA, in transmission mode, under the standard *air* atmosphere. The analyzed samples in amounts below 2 mg were grounded in the ball mill, along with calcium bromide. Then the ground powder was molded into pellets. The corresponding spectra were taken throughout the whole middle infrared region (4000–400 cm^−^^1^) and normalized by converting to unit mass. The calcium bromide (KBr, CAS: 7758-02-3) used was a FTIR-grade product purchased from Sigma-Aldrich Chemie, Steinheim, Germany. The KBr was dried at 120 °C, in a common oven, in air atmosphere, before measurement. The resolution of spectra was set to 4 cm^−1^, and the number of scans was 32 in every case.

Next, the empty *MCM-41 matrix* was exposed to various conditioning treatments that consisted of three different *drying* procedures before BDS measurements. These sample preparations of *MCM-41 matrix* included the following: (i) *drying* at specific drying temperature of 120 °C, in the VO, for different drying times, 0,1,3,6 and 7 days, followed by filling the capacitor under *air* (*O_2_ + N_2_ + moisture*) atmosphere; (ii) *drying* at 120 °C for 1 day, in the VO, followed by filling the capacitor under inert *nitrogen* atmosphere in the glove box (GB); and finally (iii) *drying* at 120 °C directly in the BDS cell, under *N*_2_ atmosphere, for different drying times, 1–8 h with subsequent immediate BDS measurements at 150 K.

### 2.2. BDS Measurements

The confined *n-PrOH/MCM-41 systems* for BDS studies were prepared by adding *n-PrOH into* the *MCM-41* powder, drop by drop, and stirring till the desired weight ratio was achieved (see Table 1). For the underfilled and saturated cases, capillary forces were allowed to fill the accessible pores of *matrices* with *n-PrOH* so that no liquid remained on the external surface *of the silica* grains. By further adding the *n-PrOH medium,* the overfilled state with the *n-PrOH molecules* in the inter-grain *silica* space was achieved. Two different environments were tested for capacitor preparation and for the *n-PrOH* filling of *MCM-41 matrix*: under regular air (*mixture of N*_2_*+O*_2_*+moisture* (*H*_2_*O*) in laboratory and under inert *N*_2_ atmosphere in the GB. The *theoretical* and *real* mass fractions of the *filler* for each *filler/confiner system*, as well as the ratios of the latter quantity to the former one, are listed in Table 1.

BDS measurements were performed by using a high-resolution Novocontrol dielectric analyzer in the range 10^−1^–10^7^ Hz. Isothermal frequency scans were performed over the temperature interval from 100 to 300 K, with a temperature step of 5 K. The *MCM-41* powder and *n-PrOH* containing *MCM-41* powder samples and liquid samples were placed between two parallel plate capacitors, without or with a Teflon spacer, with a thickness of 100 μm, respectively.

## 3. Results

### 3.1. Effect of the Conditioning of the Empty MCM-41 Matrix on the M-IR and BDS Response.

First, we investigated the influence of various thermal treatments on the empty *MCM-41 matrix* by M-IR and BDS techniques. The treatments consisted of (i) *drying* the *MCM-41 samples* at a specific drying temperature, *T*_d_ = 393 K (120 °C), for various drying times, *t*_d_, in the VO or directly in the BDS cell; and (ii) subsequent sample preparation of the variously dried *MCM-41 matrix* into the M-IR cell, under *air* or into the capacitor of the BDS cell, under various atmospheres, i.e., under regular *air* or inert *nitrogen* (*N*_2_) in the GB.

Figure 1 shows the M-IR spectra of the *MCM-41* as a function of drying time, *t*_d_ = 0, 1, 3, 6 and 7 days, in the VO, followed by sample preparation of the M-IR cell under *air* atmosphere. Here, the spectral bands at the wavenumbers of 1630 and ~3500 cm^−1^ are ascribed to the bound *H_2_O molecules* to the silanol groups of *silica* [15,16]. The presence of even a small amount of *water* at the surface of the *silica matrix* may significantly contribute to its dielectric response [19,20,21,22]. After drying the empty *MCM-41 material* in the VO for six days, followed by subsequent sample preparation of the M-IR cell under *air* atmosphere, the intensity of this particular peak becomes almost constant.

Figure 2 shows BDS spectra of the *MCM-41 matrix* samples under different drying and atmosphere conditions, at some representative temperature, *T* = 150 K. As it can be seen, the dielectric response of those samples exposed to air for capacitor preparation after six days of drying exhibits a relatively intense loss peak at f_max_ (150 K, *air*) ≈ 1 × 10^4^ Hz. In contrast, the *MCM-41* samples exposed to *N*_2_ for capacitor preparation after one day drying exhibit almost one order of magnitude lower intensity centered at a relatively lower f_max_ (150 K, *N*_2_) ≈ 3 Hz. These results indicate that the quality of atmosphere in contact with the *matrix* after drying is more determinant than the drying time on decreasing the amount of *water molecules* of the *host matrix*. Note that, according to M-IR data, the *moisture* content after one day of drying is higher than that after the three days’ one, when measured under the same air atmosphere conditions. Consistently, among those samples exposed to *N_2_* after drying that dried inside the BDS cell under *N_2_* flow shows the lowest signal, followed by that measured right after drying and preparing, and finally, that of a sample stored under *N_2_* in the GB for five days before measurement (see Figure 2). It is worth noting that the stepwise drying of the *MCM-41 silica* reaches the stationary state (no appreciable evolution of the signal) after just two cumulative hours at 393 K.

These findings emphasize the important effect of matrix thermal treatment, atmosphere exposure and sample preparation conditions on the dielectric response of the empty *MCM-41 matrix.* Sample preparation details can be critical to understand certain surface effects and should be properly indicated and underlined in the future literature.

### 3.2. Effects of Preparation of the Confined n-PrOH/MCM-41 Systems under Various Atmospheres.

Although the last special *drying* treatment of *MCM-41 matrix* directly in the BDS cell is the most effective, it is technically impossible to apply this kind of treatment for own BDS measurements of any filled *organic/MCM-41 system* because of the inevitable short-term manipulations with the *sample* under air. Therefore, we have used the standard *drying* of the *MCM-41 matrix* in the VO and investigated the effects of both basic types of atmospheres (air and *N*_2_) during preparation of the confined *n-PrOH/MCM-41 systems*. Figure 3a–c presents the BDS spectra for the various *n-PrOH/MCM-41 systems* filled to the three different filling fractions under *air* or *N*_2_ atmosphere, respectively. Both spectral data at different atmospheres exhibit some similar and some different features. This is clearly demonstrated in Figure 4 and Figure 5, together with the respective spectral responses from the empty *MCM-41 matrices* at one selected representative temperature, i.e., 180 K in both linear and logarithmic representations of dielectric loss.

A similar feature in all of the confined *n-PrOH/MCM-41 systems* is the presence of a small and broad peak (peak 1) at higher frequencies and lowest temperatures. On the other hand, significant differences consist in the character of dielectric spectra for different filling factors, regardless of the used atmosphere. For the overfilled and the saturated *n-PrOH/MCM-41 systems,* we observe a bimodal form of the respective spectra with peaks 2 and 3 at higher or lower frequency, respectively. In addition, for both of these types of confined systems differing in their filling fraction, the opposite trend in the peak intensities is evident, i.e., higher intensity of peak 2 with respect to that of peak 3, in accord with intuitive expectation. Although, in the case of the confined *samples* exposed to *air* during sample preparation, peak 2 may contain partial contribution from the losses of the bare *MCM-41 matrix*—the relative intensities of peaks 1 and 2 are of the same order as those for samples prepared under *N*_2_ atmosphere, where the dielectric loss of the *MCM-41* is minimized. On the other hand, the underfilled *n-PrOH/MCM-41 samples* exhibit broad unimodal spectra with one spectral feature (peak 3). In general, the characteristic timescales for each of the processes mentioned above are only slightly dependent on the atmosphere under which samples were prepared (pore filling and capacitor preparation).

### 3.3. Effects of the Filling Fraction in the Confined n-PrOH/MCM-41, N_2_ Systems under the Most Optimal Atmosphere Conditions.

From the previous test BDS measurements, it is evident that the most optimal confined *n-PrOH/MCM-41 system* treatment consists of *drying* the empty *MCM-41 matrix* in the VO at 120 °C for one day, with the subsequent pore and the capacitor fillings in the GB under *N*_2_ atmosphere. Figure 6 displays linear representations of the spectral evolution in all the three confined *n-PrOH/MCM-41* differing in the filling fraction. As already mentioned in the previous section, the dramatic qualitative difference between the underfilled *n-PrOH/MCM-41* and the saturated and overfilled *n-PrOH/MCM-41 ones* can be found. In contrast to logarithmic representation in Figure 3a,b, which emphases peak 1 at the lowest temperatures and the highest frequencies, linear representations of the LF-BDS spectra demonstrate more pronouncedly this difference in further peak features observed in intermediate temperatures of the BDS window. For the filling fraction 0.24, only one broad peak 3 at relatively lower frequencies exists, and in the latter ones, two peak features, i.e., peaks 3 and 2, are evident. Thus, in the saturated sample, peak 2 is relatively smaller than peak 3, and it appears over the intermediate temperature range at higher frequencies, while peak 3 is larger and situated at lower frequencies. Finally, for the overfilled system, the frequency positions of peaks 2 and 3 are similar as in the previous fully filled case, but their relative intensities are reversed. 

Dielectric spectra for the *most optimally* treated *MCM-41 matrix* and the confined *n-PrOH/MCM-41 systems* were analyzed, using an additive model with the following relaxation function for the imaginary part of the dielectric permittivity ε″(ω): ε″(ω) = −Im {Σ_i_ Δε_CC,i_/[1 + (iωτ_CC,i_)^α^_CC,i_]−iσ/ε_0_ω^s^}(1)

The first term on the right-hand side of Equation (1) describes dielectric relaxation due to reorientation of molecules related to peaks i = 1, 2 and 3. We have adopted the Cole–Cole (CC) function [7] for relaxation function for each component, where Δε_CC_,_i_ and τ_CC,i_ represent a relaxation strength and characteristic times of ith dielectric relaxation with τ_max,i_ = τ_CC,i_, respectively. The exponent α (0 ≤ α ≤ 1) is a measure of relaxation time distribution of τ_CC,i_; when α = 1, the CC function reduces to the Debye (D) function with no distribution of τ_i_. The last term is an “apparent” conductivity term accounting for dc-conductivity and additional Maxwell–Wagner–Sillar effects (MWS), due to the presence of multiple interphases.

Figure 7 summarizes the maximum relaxation times, *τ*_max_, for all the three *n-PrOH/MCM-41, N*_2_
*systems,* as obtained from the detailed spectral fitting, using Equation (1), together with the peak maximum of the main peak (the so-called Debye relaxation) of the bulk *n-PrOH*, as determined from the peak maxima. In all the bulk and confined cases, the non-Arrhenius character of the corresponding timescales of one or two main peaks (peaks 3 and 2) over the measured frequency range of the BDS technique is observed with the fitting parameters from the Vogel–Fulcher–Tamman–Hesse (VFTH) equation listed in Table 2. In Table 2, the fitting parameters from the Arrhenius equation for the secondary process (peak 1) in the confined states (data not plotted in Figure 8) are also included together with those for the bulk state from the literature [39,40,41].

Two groups of the timescales are clearly evident. The lower-frequency peak 3 appears in all the three confined underfilled, saturated and overfilled *n-PrOH/MCM-41 systems*, while in the former case, it is slower than in the remaining cases with the rather comparable timescales. On the other hand, the higher-frequency peak 2 is observable only in the saturated and overfilled *n-PrOH/MCM-41 systems,* with its frequency position lower, but quite close to that of the main peak in the bulk *n-PrOH medium*. Moreover, a slight difference does exist between the two confined cases in the spirit that the overfilled sample lies mostly in between the saturated *n-PrOH/MCM-41 system* and the bulk *n-PrOH*.

Further, short-range low-*T* extrapolations of the corresponding VFTH equations provide the *pseudo* glass transition temperatures, “*T*_g_”, operationally defined as the temperatures at which the peak maximum relaxation time reaches 100 s. In such a way determined “*T*_g_” values for two basic relaxations in the confined *n-PrOH/MCM-41*
*systems* are as follows: “*T*_g_”(peak 3) = 116 ± 1 K and “*T*_g_”(peak 2) = 101.5 ± 0.5 K. The latter value is comparable with “*T*_g_”(bulk) = 104.5 K, as determined for the main peak in the bulk *n-PrOH medium*. Note that, in the case of the bulk *n-PrOH,* the main peak consists of the dominating, slower Debye process and the smaller and faster primary α relaxation, which correlates with the calorimetric and mechanical data. Consequently, this latter process is related to the *true* glass transition temperature, *T*_g_(bulk) = 99 K, being lower than the *pseudo* glass transition temperature, “*T*_g_”(bulk) = 104.5 K [39,46,47,48].

Figure 8a,b displays two further relaxation parameters, i.e., relaxation strengths, Δε_CC,i_, and shape parameters, α_CC,i_, of processes 3 and 2, as a function of temperature. By increasing the temperature, for the underfilled sample, the relaxation strength of process 3 remains quasi-constant. On the other hand, the mutually opposite trends in the relative strengths for processes 3 and 2 in the saturated and overfilled ones are observed. In particular, the corresponding quantities for process 3 decrease, while those for the process 2 increase with elevated temperature. Finally, the shape parameter gradually increases for the underfilled system, indicating the narrowing the relaxation-time distribution. As for the saturated and overfilled samples, a similar effect is found for the lower frequency process 3 with a quasi-saturation at higher-T region for the former, but with continuing narrowing for the later approaching that for the high-frequency process 2. As for the high-frequency process 2, for both the samples, the α_CC,i_ values are more or less close and quasi-constant with an increasing temperature. 

## 4. Discussion

On the basis of these experimental findings, we suggest the existence of one dynamic phase in the underfilled *samples,* in contrast to the presence of two distinguishable dynamic ones in the saturated and overfilled *n-PrOH/MCM-41 systems* almost independently on the preparation atmosphere. These distinct dynamic phases can be ascribed as follows (see also schematic heterogeneous dynamic model of protic polar *organic* medium in polar *inorganic matrix* in Figure 9).

In the underfilled *n-PrOH/MCM-41 sample,* the protic polar *n-PrOH molecules* are localized at the polar wall of the pore surface due to their H-bonding interaction with the *silanol groups*. Moreover, according to the very recent detailed systematic work, a series of the small molecular *alkanols* ranging from *methanol* to *octanol*, also including *n-propanol*, react with these surface silanol groups of the *MCM-41 matrix*, even at room temperature, under forming the corresponding polar *alkoxy groups* [49]. Thus, the true *interface* appears to contain three components, i.e., the rest adsorbed *H*_2_*O molecules,* the intermolecularly *H*-bonded *n-PrOH ones* and the *alkoxy groups* to the *silica* surface. As our filling fraction of 0.24 exceeds the monomolecular layer case, i.e., true *interface* sometimes called the contact layer, also the *adjacent* layers of the *n-PrOH molecules* named *interphase* contribute to the dielectric response. This formation of one or a few layers adjacent to the contact layer is supported by the amphiphilic character of the *n-PrOH molecules*, where the polar *hydroxyl* group –*OH* of the *n-PrOH* medium are attached directly via *H*-bonding to the polar *silanol* groups and even covalently via *alkoxy groups* ≡ *SiO(CH*_2_)_2_*CH*_3_. On the other side, the apolar *propyl*-*CH*_2_*CH*_2_*CH*_3_ parts of the *n-PrOH molecules* are directed into the pore, where it can preferentially interact with the same *propyl* groups from the first adjacent layer. This probably gave rise to the more denser *interphase* region with the higher orientational order which can be reduced in direction toward the central part of the pore of the *MCM-41 matrix*. In addition to the overwhelming amount of *n-PrOH medium*, the inevitable remaining *H*_2_*O molecules* from the used drying procedure under *N*_2_ atmosphere, as well as the potential *alkoxy groups*, also contribute, to a minor extent, to the total spectrum, as evidenced from Figure 2 and Figure 5.

Next, for the almost saturated *n-PrOH/MCM-41* case, besides this combined *interface* and *interphase* slower regions, the additional faster dynamic component in the BDS spectra is attributed to the more distant *n-PrOH molecules* from the pore surface which are localized in the central part, i.e., the so-called “core” region of the pore of the *MCM-41 matrix*. As it is seen from Figure 7, the *n-PrOH molecules* in this central “core” region exhibit a somewhat slower dynamics compared to that in the bulk *n-PrOH medium*, so that they exhibit the *bulk*-like behavior.

Finally, in the overfilled *n-PrOH/MCM-41 system* with f = 0.46, an exceeding part of the *n-PrOH medium* over the saturated filling fraction 0.39 must be situated outside of the *MCM-41 grains*. Consequently, while the dynamics of the *n-PrOH* stemming from the confined part of the *n-PrOH* in the intrapores is rather similar to that from the saturated case, the dynamics of the exceeding part of the *n-PrOH filler* is a bit faster compared to the fully filled case, but still a bit slower than in the bulk *n-PrOH medium*. This is also due to some contribution from the *n-PrOH molecules* in the free space between the *MCM-41 grains* which are partially bonded to the outer surface of the *MCM-41 grains.* As the outer grain surface in the *MCM-41 matrix* is essentially smaller than the inner pore surface within the pores of the *MCM-41 grains*, the bounded *n-PrOH* in this outer *interface* (and eventually some *interphase*) will contribute in significantly smaller extent, so that the higher relative intensity of peak 2 compared to peak 3 stems from the *n-PrOH medium* in the intergrain (interpore) spaces with the *bulk*-like behavior (Figure 5 and Figure 8).

The fact that process 2 is narrow and its shape does not change appreciably with temperature supports the interpretation of this process as originated by *bulk*-like *n-PrOH* molecules. Concerning the relative strength of processes 2 and 3 and their temperature dependence, it is consistent with the expectation that the number of molecules behaving as *bulk*-like increases with temperature due to an increase of mobility and a decrease of the size or number of cooperating molecules. Consistently, the strength of process 3 in the underfilled sample remains more or less constant, as in this sample, molecules just belong to the first monolayer *interface* and to *interphase,* and their number is too low to switch to a *bulk*-like behavior as temperature increases. In line with these observations, the *T*-dependence of the characteristic times of process 2 is more *“fragile”,* and process 3 becomes *“stronger”* the lower the *n-PrOH* content.

In connection with the aforementioned findings and their interpretation in terms of distinct dynamic phases, it is of interest to see how the BDS spectra of the saturated *n-PrOH/MCM-41, N*_2_
*sample* evolves during simple gradual thermal treatment at 300 K, followed by that at 310 K, as performed directly in the BDS cell, i.e., during “drying”, the confined *n-PrOH/MCM-41,N*_2_
*system* due to the release of the *n-PrOH medium.*
Figure 10a shows the spectral changes for the saturated *sample* first cooled and measured at the selected temperature of 175 K (scan 1) and after two heatings that followed, for seven minutes, at 300 K. While peak 3 remains the same during all the heating cycles (scans 2 and 3), peak 2 at almost the same position decreased due to evaporation of the *n-PrOH medium* from the central region of the pore of the *MCM-41 matrix,* where the *n-PrOH molecules* interact mutually by the weaker intermolecular forces than the *ones* bounded directly to and being more adjacent to the polar surface wall of the pores of the *MCM-41 matrix*. Comparison with the first scan of the overfilled *n-PrOH/MCM-41, N*_2_
*sample* (Figure 7) shows that the position of peak 3 is the same for both *samples.* In contrast, peak 2 is more intense, indicating somewhat looser dynamics for this process in the overfilled *sample* compared to the similar peak 2 in the saturated *sample*. This would be evidence of the excess *n-PrOH medium* localized outside the pores of the *MCM-41 matrix*. Next, the temperature of this partially dried *sample* was increased toward 310 K, and heating and cooling cycles were repeated 17 times, with subsequent detection of the spectra at 175 K (see Figure 10b). As it can be seen, peak 2 is gradually fully eliminated, and peak 3 is also gradually reduced, with a slight shift toward the lower frequencies. Thus, all these observations appear to be consistent with the picture of two distinct dynamic phases and support the proposed schematic model in Figure 9.

## 5. Conclusions

We presented a detailed investigation of the relaxation dynamics in the representative of short-chain *n-alkanols,* namely *n-propanol* confined in the regular *MCM-41 matrix* as a function of a series of external parameters by low-frequency dielectric spectroscopy. These external parameters concerned the following three issues: (i) conditioning the empty *MCM-41 matrix* and its subsequent filling into the capacitor under various atmospheres, i.e., *air* or *N*_2_; (ii) conditioning the empty *MCM-41 matrix* and subsequent filling the *n-PrOH medium* into the capacitor under various atmospheres; and finally (iii) variation of the filling factor for the most optimally treated *MCM-41 matrix.* In the first case, relatively large sensitivity of the BDS response of the empty *MCM-41* to its various atmospheres, especially at the capacitor filling, was found. In the second one, in spite of this, the dynamics are not very essentially affected by the used environment at the pores, as well as at the capacitor filling. Finally, the filling factor has a paramount impact on the spectral form of the BDS response, indicating the spatial heterogeneity of the dynamics of the confined *n-PrOH* in *MCM-41 matrix* due to the presence of the two distinct dynamic phases.

## Figures and Tables

**Figure 1 polymers-12-00610-f001:**
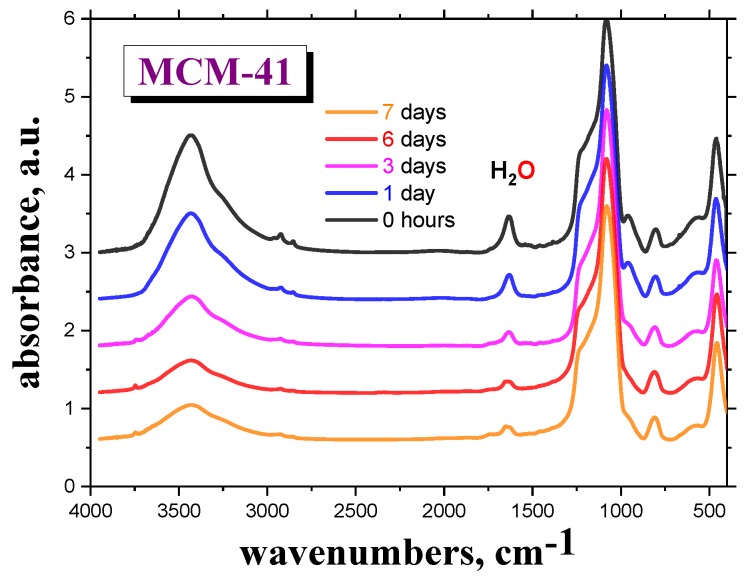
M-IR spectra of the empty *MCM-41* dried in the VO at specific drying temperature, *T*_d_ = 120 °C, for various drying times, *t*_d_ = 0, 1, 3, 6 and 7 days. Filling the M-IR cell was carried out at *RT*, under *air* atmosphere. The peaks at ν~1630 and ~3500 cm^−1^ belong to the bonded *H_2_O* to the silanol groups of *MCM-41 matrix* [15,16].

**Figure 2 polymers-12-00610-f002:**
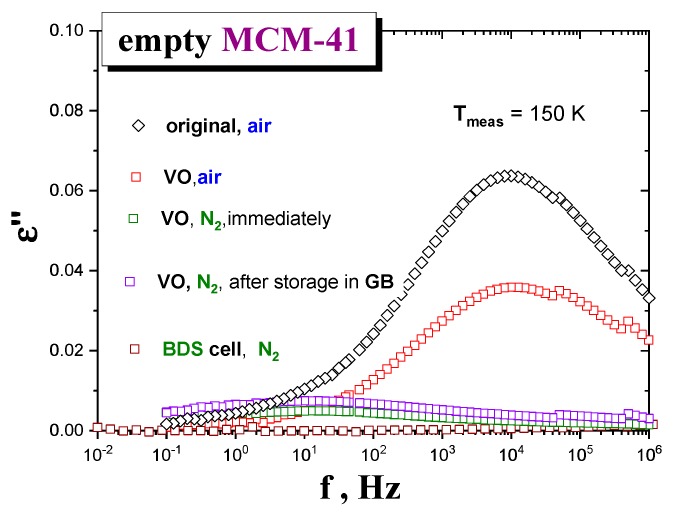
Comparison of the BDS spectra recorded at 150 K of the empty *MCM-41* in the original (untreated) state (black diamonds) and in a series of thermally treated empty *MCM-41* samples by drying and subsequent manipulation, i.e., loading into the capacitor by various ways: (**a**) dried in the VO at 120 °C for six days and subsequently filled into the capacitor under the *air* (*N*_2_
*+ O*_2_
*+ moisture*) atmosphere (red squares); (**b**) dried in the VO at 120 °C for one day and subsequently filled into the capacitor under *N*_2_ in the GB and measured (i) immediately (green squares) and (ii) after five days’ storage in the GB (blue squares) and finally; (**c**) dried at 120 °C for 2 h, directly in the BDS cell, under *N*_2_ (brown squares). All the empty *MCM-41 samples* were undergone by the same cooling treatment with −2 K/min from *RT* down to 100 K and subsequently measured isothermally with Δ*T* = 5 K from 100 up to 300 K.

**Figure 3 polymers-12-00610-f003:**
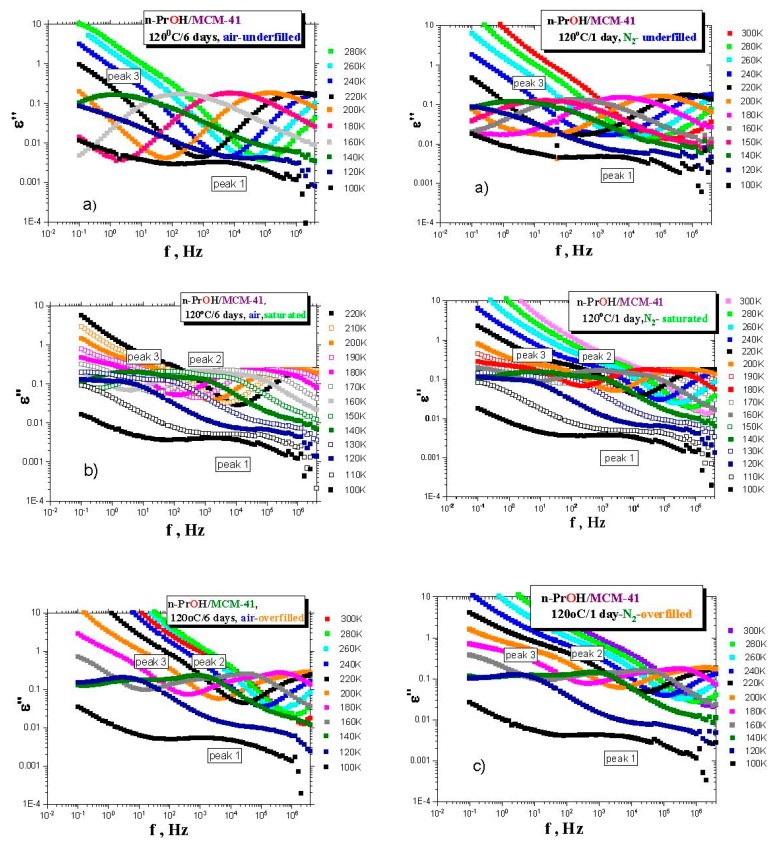
BDS spectral evolutions for the confined *n-PrOH*/*MCM-41* underfilled (**a**), saturated (**b**) and overfilled (**c**) *systems* prepared by *drying* of the *MCM-41* at 120 °C for six days in the VO and subsequent loading of the *n-PrOH medium* into the capacitor under air (left column) and *drying* of the *MCM-41* at 120 °C for one day in the VO and subsequent loading of the *n-PrOH medium* into the capacitor under *N*_2_ (right column) The samples were cooled with −2 K/min from *RT* down to 100 K and subsequently, isothermally heated with Δ*T* = 5 K from 100 K up to 280–300 K.

**Figure 4 polymers-12-00610-f004:**
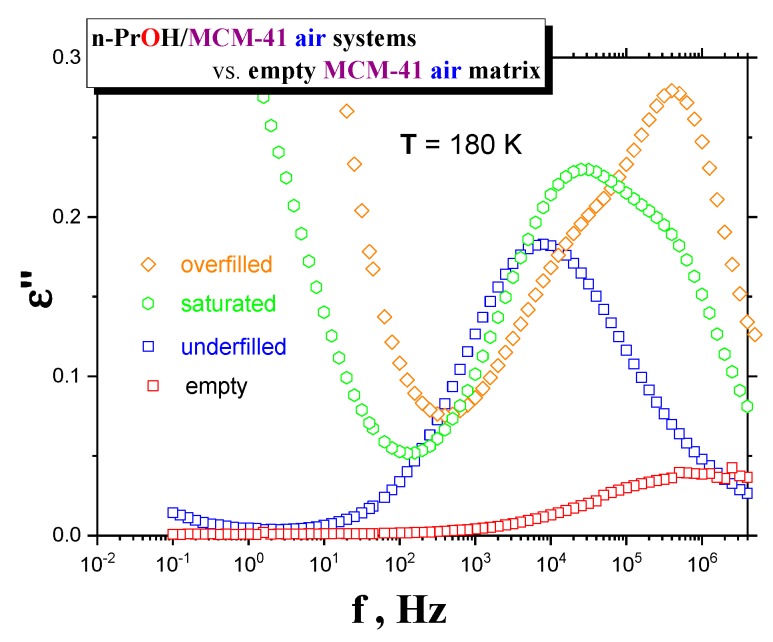
Comparison of the BDS spectra at 180 K of empty *MCM-41 matrix* vs. underfilled, saturated and overfilled *n-PrOH/MCM-41*, *air systems* after cooling with −2 K/min from *RT* down to 100 K and subsequently isothermal measured with Δ*T* = 5 K step up to 300 K.

**Figure 5 polymers-12-00610-f005:**
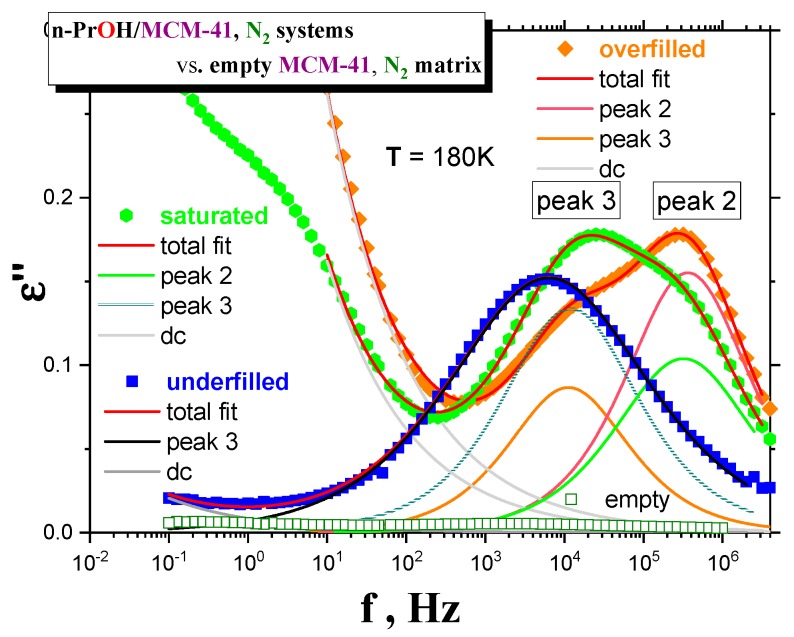
Comparison of the BDS spectra at 180 K for underfilled, saturated and overfilled *n-PrOH/MCM-41, N*_2_
*systems,* as well as for empty *MCM-41, N*_2_ and *n-PrOH* after cooling with −2 K/min from *RT* down to 100 K and subsequently isothermal heated with Δ*T* = 5 K step up to 300 K. Spectral fits using Equation (1) are included.

**Figure 6 polymers-12-00610-f006:**
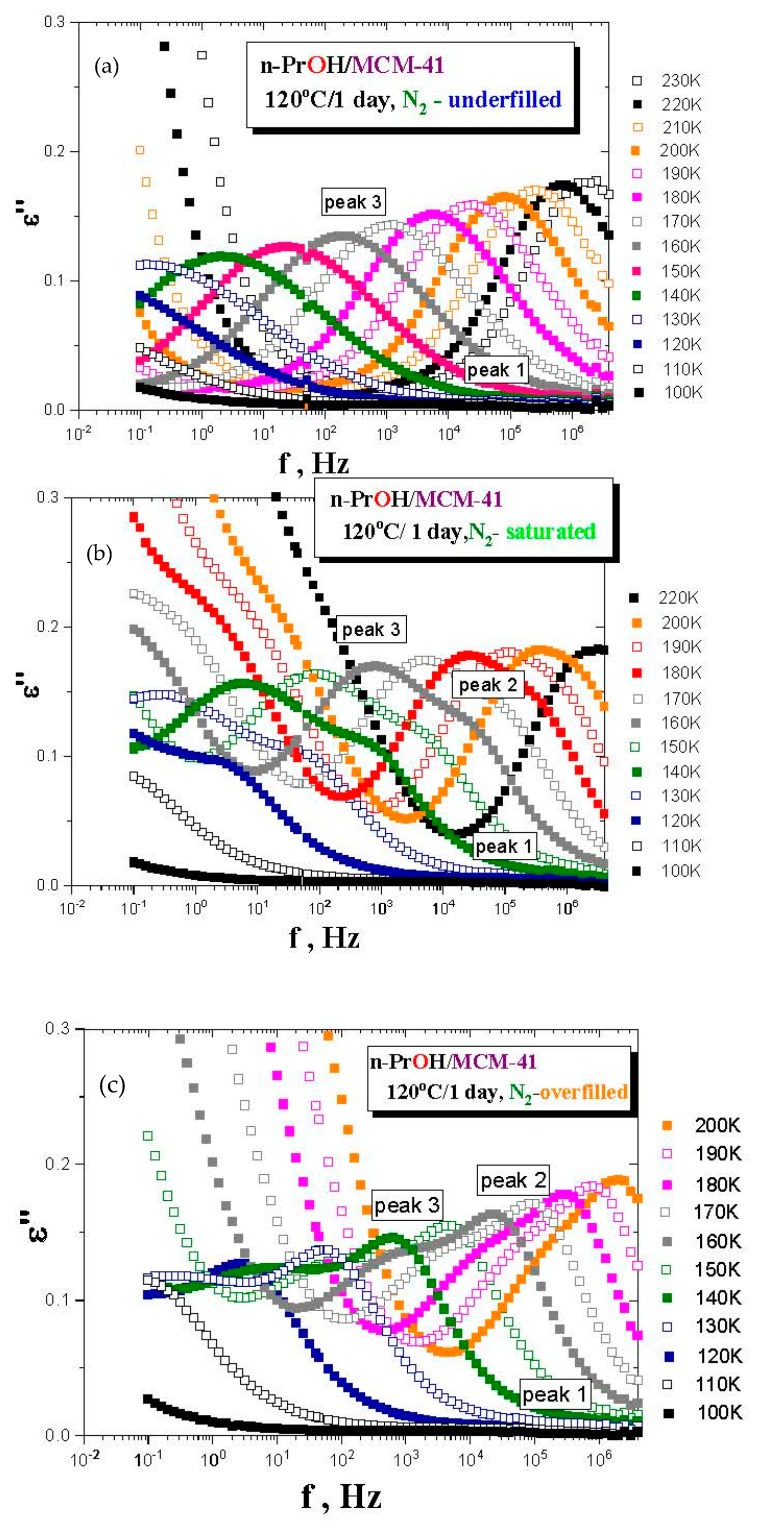
Temperature evolution in the confined *n-PrOH/MCM-41, N*_2_
*systems*: underfilled (**a**), saturated (**b**) and overfilled (**c**), up to 230 and 200 K for the first two or the last filling factor.

**Figure 7 polymers-12-00610-f007:**
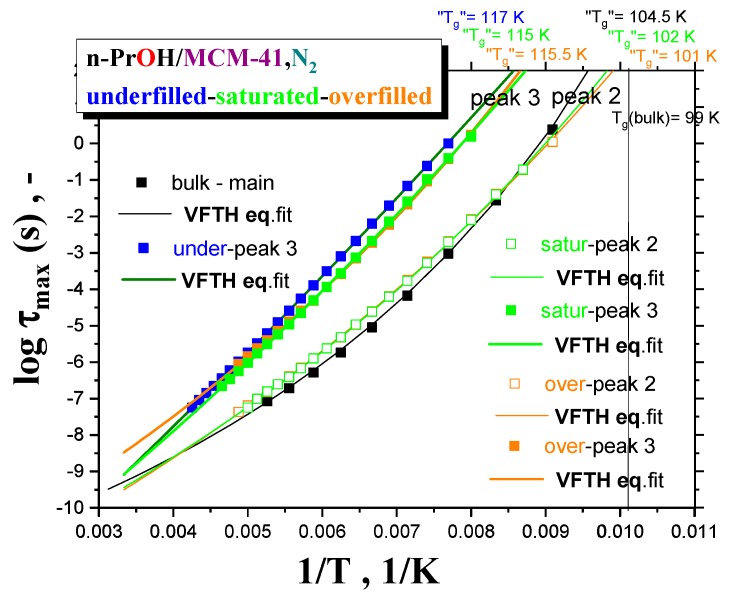
Arrhenius plot of the maximum relaxation times in all the three variously filled *n-PrOH/MCM-41, N_2_ systems,* determined from the unimodal spectra for the underfilled *n-PrOH/MCM-41, N*_2_
*system* and from the bimodal ones for the saturated and overfilled *n-PrOH/MCM-41, N*_2_
*ones,* together with the corresponding VFTH equation fits with parameters in Table 2.

**Figure 8 polymers-12-00610-f008:**
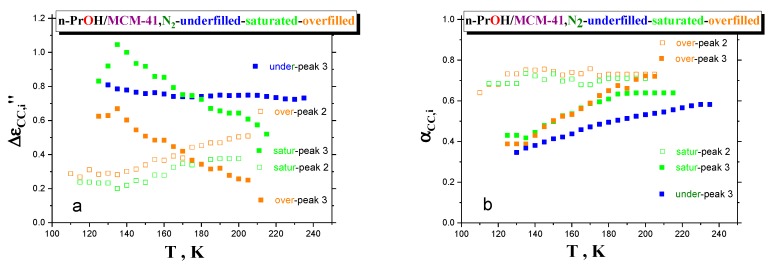
Temperature dependences of the relaxation strength (**a**) and shape parameter (**b**) for all the three variously filled confined *n-PrOH/MCM-41, N_2_ systems.*

**Figure 9 polymers-12-00610-f009:**
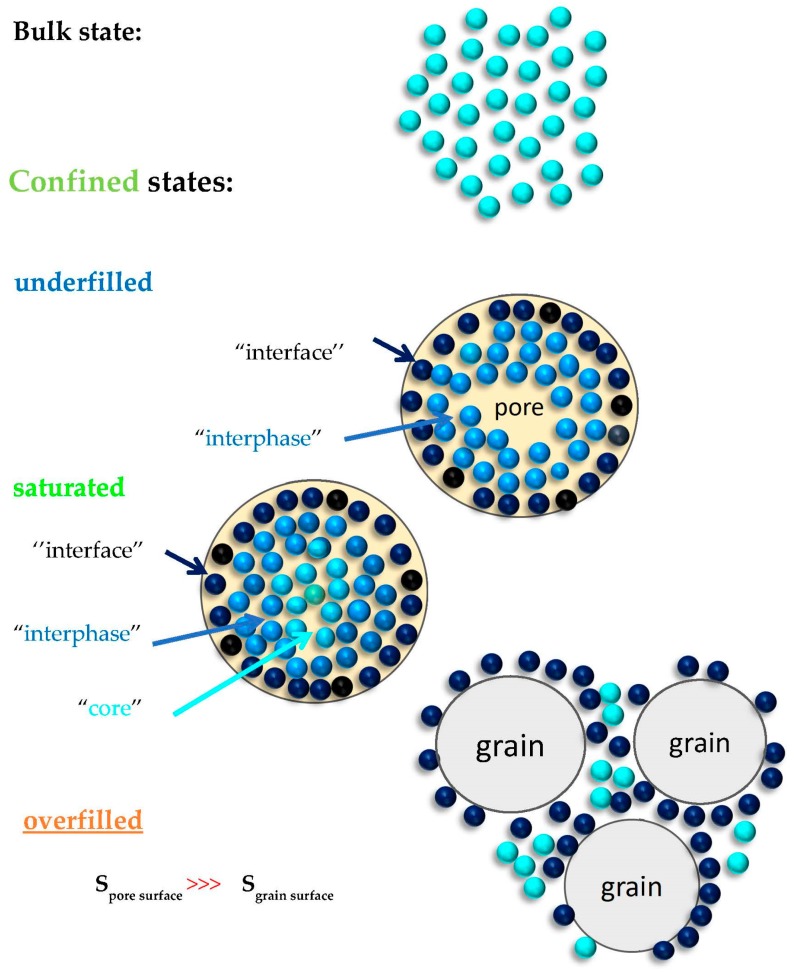
Schematic heterogeneous dynamic model of protic polar *organic* medium in polar *inorganic matrix.* Three basic structural-dynamic features in three confined *n-PrOH/MCM-41 systems* are as follows: (i) “*interface*” formed by the monomolecular layer of the polar *molecules* of *n-PrOH medium,* the rest polar *H*_2_*O molecules* directly anchored to the polar ≡ *SiOH groups* at the pore surface of *MCM-41 matrix,* as well as by the possible *alkoxy-groups*; (ii) “*interphase*”, i.e., quasi-ordered layer of the amphiphilic polar *molecules* of *n-PrOH medium*; and (iii) “core”, i.e., the *bulk*-like polar molecules of *n-PrOH medium* at the central part of pore of the *MCM-41 matrix* (see the text). Note that, while in the first two underfilled and saturated cases the pore vs. *n-PrOH medium particle* sizes relationships are depicted more or less realistically from the mutual size viewpoint, in the overfilled one, the mutual relationships between the *n-PrOH medium molecules* and the grain sizes are oversimplified due to the essential difference between their dimensions: *D*_grain_(a few μms) >>> *D*_n-PrOH_^vdW,eq^ (a few nms).

**Figure 10 polymers-12-00610-f010:**
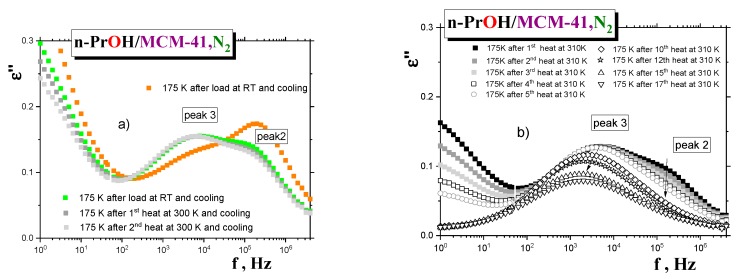
Spectral evolution of the originally saturated *n-PrOH/MCM-41, N*_2_
*system,* after two heating-cooling cycles at 300 K (**a**) and after following seventeen-fold heating-cooling ones at 310 K (**b**). BDS spectrum for the overfilled *sample* is added for comparison.

**Table 1 polymers-12-00610-t001:** Matrix and confined n-PrOH/MCM-41 systems.

Matrix	Pore Diameter *D*_pore_, Å	Pore Volume *V*_pore_, cm^3^/g	Pore Area *S*_pore_, m^2^/g	*F*_n-PrOH,theo_ *	*F*_n-PrOH,real_ **	*X **** %
*MCM-41* underfilled	40	0.80	1098	0.39	0.235	60.3
*MCM-41* saturated	40	0.80	1098	0.39	0.359	92.1
*MCM-41* overfilled	40	0.80	1098	0.39	0.456	116.9

** F*_n-PrOH,theo_ = *m*_n-PrOH_/(*m*_n-PrOH_*+m*_MCM-41_), the theoretical mass fraction of *n-PrOH* medium with respect to the *n-PrOH/MCM-41* system estimated using the density of *n-PrOH* at room temperature, *ρ*_n-PrOH_(RT) = 0.803 g/cm^3^ under the complete accessibility condition of all the regular pores for the *n-PrOH* medium, ** *F*_n-PrOH,sat_ = *m*_n-PrOH_/(*m*_n-PrOH_+*m*_MCM-41_), the real experimental mass fraction of *n-PrOH* in the *n-PrOH/MCM-41* system corresponding to the three different filling situations of the *n-PrOH* in the *n-PrOH/MCM-41*system, *** *X* = (*F*_n-PrOH,sat_/*F*_n-PrOH,theo_) × 100%.

**Table 2 polymers-12-00610-t002:** Fitting parameters from the VFTH equation: *τ*_max_ = *τ_∞_*exp[*B/*(*T−T*_0_)] for the main peak features (peaks 3 and 2) and the Arrhenius equation: *τ*_max_ = *τ_∞_*exp[*B/T*] for the secondary peak one (peak 1) in the bulk *n-PrOH* and the confined *n-PrOH/MCM-41, N_2_ systems.*

System	Peak	Δ*T* (K)	log *τ_∞_*	*B* (K)	*T*_0_ (K)
Bulk	1	77–121	−14.2	1075	0
	2	110–190	−11.9	630.3	59
Confined					
underfilled	1	100–125	−16.1	1208.1	0
	3	130–235	−15.4	1801.7	12.9
saturated	1	100–140	−20.1	1803.9	0
	2	115–200	−12.9	912.8	40.7
	3	125–215	−14.4	1468.5	25.1
overfilled	1	100–140	−14.8	1094.3	0
	2	110–205	−13.2	982.5	36.4
	3	125–205	−12.6	1050	43.1

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
