# Peer review of "Dynamics of Confined Short-Chain alkanol in MCM-41 by Dielectric Spectroscopy: Effects of matrix and system Treatments and Filling Factor"

_polymers, 2020, doi:10.3390/polym12030610_

Round 1
Reviewer 1 Report
The manuscript (polymers-687738) of J.Bartos et al. on „Dynamics of confined short-chain alkanol in MCM-41 by BDS; Effects of matrix and system treatments and filling factor“ deals with the glassy dynamics of anhydrous n-Propanol (n-PrOH), an organic glass contained in the (MCM-41) silica matrix as measured by Broadband Dielectric Spectroscopy (BDS). Special emphasis is given tot he impact oft he filling factor and aspects of preparation. This is certainly an interesting topic albeit the many studies which exist already on this topic. The main conclusion of the authors „Finally, various filling factors of n-PrOH medium in the optimally treated MCM-41 lead to unimodal or bimodal spectra being interpreted in terms of the two distinct dynamic phases in the confined states“ (s. abstract) is not surprising and in qualitative accord with previous studies. In detail the following topics have tob e improved.
I miss especially a quantitative analysis of the relaxation processes in which also the dielectric strength is determined. This is nowadays standard and demonstrated fort he polymer Polyispbutylether (PIVBE) contained in MCM 41 in a textbook edited in 2003. The pore surface of MCM41 is well known and based on the dielectric strength one can determine the relative contributions of the two processes and how they change with temperaure. MCM´s are obtained as microcrystals, hence their surface can be – at least estimated. From that it should be possible to determine the number of molecules interacting with the surface oft he „grains“ (s. Fig.9). The existence of „interphase- like molecules as depicted in Fig 9 „saturated“ is in my view not at all proven by the present data. 2 is a nice example of the impact of the preparative conditions. But this is really not new. In the title of a publication one should avoid abbreviations; I doubt that every reader of „Polymer“ deciphers „BDS“. In the references important publications and books are missing.In summary this paper needs a thorough revision and especially a quantitative analysis of the dielectric spectra, before it can be recommended for publication.
Reviewer 2 Report
Dielectric spectroscopy, as an important characterization tool, has been widely applied for studying the physical, chemical, and biological properties and changes of matters. In this manuscript, Bartoš et al. carried out detailed study of MCM-41 using broadband dielectric spectroscopy (BDS). The structure and the representation of the paper is good. I only have a few minor remarks:
Please describe the calibration methods of the broadband dielectric spectroscopy for the liquid and power samples. The axis titles of figures are suggested to be centered.Author Response
Please see the attachment.

Reviewer 3 Report
The main achievement of the work is the study of various treatments of MCM-41 matrix and investigated system. The authors examined the effect of saturation on molecular dynamics.
The investigations of confined materials seem to have great importance in present-day physical chemistry and many other scientific and industrial areas.
The scientific target is interesting, but the manuscript needs to be improved.
Did the authors evaluate saturation (underfilled/saturated/overfilled) of the sample by eye?
Separated absorption peaks ascribed to relaxation processes should be presented in figures 4 and 5.
Figure 5: What is the origin of the process in the conductivity region for the saturated sample?
Figure 6: Do the determined results follow Arrhenius or VFTH law? The fitting line is needed.
Table II: Why is T0 equal 0 for “peak 1”.
The parameters of VFTH equation should be explained in the text.
What is the difference between “true” and “pseudo” glass transition? The short discussion with references must be addressed in the manuscript.
I have found some shortcomings:
Figure 3: “oC”; the panels should be listed without doubling the same indicator.
You do not need to repeat the full names of experimental techniques (e.g. line 56) when the abbreviations were introduced beforehand.
Please check the numeration of references.
Round 2
Reviewer 1 Report
The revised version of this manuscript is now recommended for publication in "Polymers".
Reviewer 3 Report
The authors addressed most of the remarks.
Most important, the authors improved the analysis of BDS data.
The introduction misses some of the recent results in the field.
However, this is not the review article and it seems acceptable.
The revised manuscript seems to be valuable and can be interesting for the readers.